# Reliability Evaluation of Fan-Out Type 3D Packaging-On-Packaging

**DOI:** 10.3390/mi12030295

**Published:** 2021-03-10

**Authors:** Pao-Hsiung Wang, Yu-Wei Huang, Kuo-Ning Chiang

**Affiliations:** Department of Power Mechanical Engineering, National Tsing Hua University, Hsinchu 300, Taiwan; s101033812@gmail.com (P.-H.W.); s106033852@m106.nthu.edu.tw (Y.-W.H.)

**Keywords:** fan-out wafer-level package, finite element, glass substrate, reliability life, packaging-on-packaging

## Abstract

The development of fan-out packaging technology for fine-pitch and high-pin-count applications is a hot topic in semiconductor research. To reduce the package footprint and improve system performance, many applications have adopted packaging-on-packaging (PoP) architecture. Given its inherent characteristics, glass is a good material for high-speed transmission applications. Therefore, this study proposes a fan-out wafer-level packaging (FO-WLP) with glass substrate-type PoP. The reliability life of the proposed FO-WLP was evaluated under thermal cycling conditions through finite element simulations and empirical calculations. Considering the simulation processing time and consistency with the experimentally obtained mean time to failure (MTTF) of the packaging, both two- and three-dimensional finite element models were developed with appropriate mechanical theories, and were verified to have similar MTTFs. Next, the FO-WLP structure was optimized by simulating various design parameters. The coefficient of thermal expansion of the glass substrate exerted the strongest effect on the reliability life under thermal cycling loading. In addition, the upper and lower pad thicknesses and the buffer layer thickness significantly affected the reliability life of both the FO-WLP and the FO-WLP-type PoP.

## 1. Introduction

As electronics technology progresses, 3D integrated systems are becoming increasingly important in the realization of lightweight devices with higher performance and better miniaturization. These systems not only effectively reduce the package footprint and weight, but also improve system performance by reducing the system circuit communication length. The fan-out package structure has good electrical and thermal performance and is used in many applications for system integration. Moreover, packaging-on-packaging (PoP) technology allows packages to be stacked three-dimensionally, thus achieving high-density integration and improving chip-to-chip performance (e.g., in applications with high-frequency data exchange between application processes and memory) [1,2,3].

Glass is an insulating material, and hence its electrical characteristics are more favorable than those of silicon. In addition, the thickness of a glass substrate can be modified without additional thinning and polishing processes, and through glass via (TGV) technology does not require an additional barrier layer, greatly reducing the production cost. Moreover, the coefficient of thermal expansion (CTE) of glass can be optimized to reduce warpage [4,5] and improve the reliability life of stacked fan-out wafer-level packaging (FO-WLP). Hence, a fan-out structure with a glass substrate is favorable for high-frequency applications [4,6]. However, before mass production, packaging must pass reliability life testing under thermal cycling loading; in the JEDEC(Joint Electron Device Engineering Council) standard (JESD22-A104D), the thermal range is −40 to 125 °C. Finite element analysis is widely used to optimize package structures [7,8,9,10,11,12]. In this study, we proposed an FO-WLP with a glass substrate architecture and used it as a PoP. We fabricated FO-WLP test samples, subjected them to onboard thermal cycling testing (OBTCT), and verified the results against those of the finite element models described below. In the fabricated FO-WLP structure, the corner solder joint is the critical failure point because during heating and cooling, stress and strain accumulate in these joints due to mismatches in the CTE of the package and the printed circuit board (PCB), eventually leading to failure.

We further established two-dimensional (2D) and three-dimensional (3D) models and verified their consistency in terms of the mean-time-to-failure (MTTF). To reduce computing time, the 2D model was used to optimize the reliability life by varying the upper and lower pad diameters, the CTE of the glass substrate, and the thickness of the buffer layer. Parametric studies revealed that the reliability life of the optimized FO-WLP and PoP exceeds 1000 cycles, satisfying JEDEC condition G.

## 2. Materials and Methods

### 2.1. Shape Prediction of the Reflowed Solder Joint

The geometry profile of the solder joints strongly affects the reliability life of a package. Therefore, before analyzing the reliability of the solder joints, the shape of the solder balls must be accurately described. In this study, Surface Evolver [13,14,15] software based on the energy method was used to describe and predict the solder joint shape.

When a liquid reaches static equilibrium, its total energy tends to be the lowest and its surface area the smallest. The energy of a liquid mainly comprises surface tension energy, gravitational energy, and external energy. From the total energy (Equation (1)), the restoring force in the direction of gravity can be calculated, and the shape and height of the solder ball can be estimated:(1)$$\delta E_{total}=T_{S}\iint_{S}^{}\left(div\vec{h}-\vec{n}\cdot D\vec{h}\cdot \vec{n}\right)dA+\rho g\iint_{S}^{}\left(div\frac{z}{2}\vec{k}\right)\vec{h}-\mathrm{curl}\left(\vec{h}\times \frac{z}{2}\vec{k}\right)\cdot dA-P\iint_{S}^{}\vec{h}\cdot dA$$

If Equation (1) is differentiated once, the restoring force of the solder ball can be expressed as follows:(2)$$F_{r}=\frac{\partial E_{total}}{\partial H}=\frac{\partial E_{surface\;tension}+\partial E_{gravity}+\partial E_{external\;force}}{\partial H}$$
where $E_{total}$ is the total energy related to the height of the solder ball *H*, *T_s_* is the surface tension of the solder ball, *P* is the pressure caused by the external force, *A* is the surface area of a single element on the solder ball surface, *z* is the height of the solder ball of a single element surface parallel to the direction of gravity, $\vec{k}$ represents the unit vector in the direction of gravity, $\vec{n}$ is the unit vector along the positive direction of the element surface, *F_r_* is the restoring force, ρ is the density of the solder ball, *g* is the acceleration due to gravity, and *V* is the volume of the solder ball. Furthermore, $\vec{h}$ is the perturbation equation:(3)$$\vec{h}=\left[\left(z_{top}-z\right)/\left(z_{top}-z_{base}-H\right)\right]\vec{k}$$
where $z_{top}$ and $z_{base}$, respectively, represent the upper and lower boundary conditions of the solder ball (Figure 1). By applying a slight upward or downward interference on the pad, the restoring force of the solder ball in the direction of gravity can be determined. When the restoring force of the solder ball equals the gravity exerted on the solder ball, the molten solder ball achieves static equilibrium. Then, the height and geometry of the solder ball under static equilibrium can be determined.

### 2.2. Life Prediction of the Solder Joints

In electronic packages subjected to accelerated thermal cycling tests, CTE mismatch between the package and the PCB can cause excessive stress and strain to accumulate in the solder joint with the largest distance from the neutral point (DNP); this may cause the first failure of the packaging. The Coffin–Manson strain-based empirical model is widely used to estimate the fatigue life of solder joints, and the empirical equation [16,17,18] is as follows:(4)$$N_{f}=C\;{\left(\varepsilon_{eq}^{in}\right)}^{-\eta }$$
where $N_{f}$ is the mean time to failure (MTTF), and *C* and η are material constants, usually obtained experimentally. In our model and for SAC305 solder material, these are 0.235 and 1.75, respectively [10,11,12]. The incremental equivalent inelastic strain in each temperature cycle, $\varepsilon_{eq}^{in}$, is defined as follows:(5)$$\Delta \varepsilon_{eq.}^{in}=\frac{\sqrt{2}}{3}{\left[{\left(\Delta \varepsilon_{x}^{in}-\Delta \varepsilon_{y}^{in}\right)}^{2}+{\left(\Delta \varepsilon_{y}^{in}-\Delta \varepsilon_{z}^{in}\right)}^{2}+{\left(\Delta \varepsilon_{z}^{in}-\Delta \varepsilon_{x}^{in}\right)}^{2}+\frac{3}{2}\left(\Delta \gamma_{xy}^{in}{}^{2}+\Delta \gamma_{yz}^{on}{}^{2}+\Delta \gamma_{zx}^{in}{}^{2}\right)\right]}^{\frac{1}{2}}$$
where $\Delta \varepsilon_{x}^{in}$, $\Delta \varepsilon_{y}^{in}$, $\Delta \varepsilon_{z}^{in}$, $\Delta \gamma_{xy}^{in}$, $\Delta \gamma_{yz}^{in}$, and $\Delta \gamma_{zx}^{in}$ are the incremental inelastic strains in the *x, y*, and *z* directions and the incremental inelastic shear strain in the *xy, yz*, and *zx* directions, respectively.

In this study, we used finite element simulation and the Coffin–Mason empirical model to evaluate the reliability life of the solder joint with the largest DNP.

## 3. Test Vehicle Structure and Thermal Cycling

### 3.1. Structure of the Test Vehicle

The target stackable FO-WLP was a cavity-down chip mounted on a glass interposer using TGV and redistribution lines. These chips can be stacked and molded to form PoP-type packaging using through molding via (TMV). The test vehicle [19] was a simplified FO-WLP (a 10 mm × 10 mm × 0.1 mm chip) assembled on a glass substrate of size 14 mm × 14 mm × 0.1 mm using a die attach film (Nitto, EM700). The chip and substrate were covered by a molding compound of size 14 mm × 14 mm × 0.2 mm to form the package. Next, the package was subjected to mechanical debonding (Figure 2). The underside of the glass substrate contained 432 solder joints arranged in a peripheral layout and connected in series. The diameter and pitch of the solder ball were 250 mm and 400 µm, respectively. The test board dimensions were 77 mm × 132 mm, following the JEDEC (JESD22-B111) design rule. The dimensions of all of the components are listed in Table 1. In the test board, a daisy-chain structure was used to check the electrical resistance of the chained solder joints (Figure 3). The test vehicle was deemed to have failed if its daisy-chain resistance was infinite.

### 3.2. Thermal Cycling and Weibull Distribution

For reliability testing, we subjected 16 test vehicles to onboard thermal cycling. The test conditions follow the JEDEC standard (condition G): temperature range, −40 to 125 °C; ramp rate, 16.5 °C/min; dwell time, 10 min. The Weibull distribution of the test results (Figure 4) revealed that the mean time to failure was 249 cycles when the cumulative failure percentage equaled 63.2%. Moreover, all 16 samples failed at the top of the outermost solder joint.

## 4. Finite Element Analysis of FO-WLP

We established a 2D diagonal semi-symmetric plane strain model and a 3D quarter model to study the thermomechanical behavior of the solder joints in the FO-WLP structure. As the 2D model had fewer nodes than the 3D model, it required less simulation time. However, because plane strain was assumed in the 2D model, it cannot truly represent the actual state of the test vehicle (e.g., the semi-spherical nature of the solder ball). Therefore, a 3D model closer in geometry to the actual vehicle was established to verify the robustness of the 2D model.

After verifying the finite element simulations against the experimental results, we added a TMV component to the FO-WLP structure to make it stackable. The stacked PoP architecture was evaluated by simulation under the JEDEC standard (condition G) testing condition.

### 4.1. Material Parameters

The test vehicle was composed of an Si chip, a die attach film, a glass substrate, a stress buffer layer (SBL), copper, SAC305 solder balls, a PCB, and a molding compound. We used three types of glass substrates whose CTE ranged from 3.17 to 9.8 ppm/°C and whose modulus ranged from 64 to 74 GPa, with little difference in Poisson’s ratio. The material parameters are listed in Table 2. The temperature-dependent characteristics of all of the materials were linear, except for the SAC305 solder balls. Figure 5 shows the stress–strain curve of the solder balls at different temperatures [20].

To model the creep behavior [7] of the solder balls, we adopted the Garofalo–Arrhenius hyperbolic sine model (Equation (6)), which has been widely used to simulate the thermal cycle load of solder joints:(6)$$\frac{d\varepsilon }{dt}=A\left(sinh{\left(B\sigma \right)}^{n}exp\left(\frac{-Q}{RT}\right)\right)$$
where *ε* is the strain, *σ* is the stress, *A* and *B* are the material constants, *R* is the gas constant, *T* is the absolute temperature, *Q* is the activation energy, and *n* is the stress index. Table 3 lists the values of these constants as used in this study [13].

### 4.2. 2D Plane Strain Model

For the finite element analysis of the 2D model, we assumed plane strain and used PLANE42 and PLANE182 (ANSYS, 2020R2). Surface Evolver was used to define the shape of the solder joints. Figure 6 depicts the top view of the test vehicle, where the solder joints located in the periphery array format are marked in white. The finite element model was a 2D half model built along the diagonal of the test vehicle (Figure 7). Its position is the yellow line in Figure 6. Figure 8 illustrates the finite element model of the stacked FO-WLP (PoP) structure. The upper and lower stacks of the package were connected through soldering and TMV. The TMV was 0.25 mm in diameter and located atop the solder balls.

### 4.3. 3D Quarter Symmetry Model

Unlike a 3D finite element model, a 2D plane strain model cannot accurately represent semi-spherical-type features. However, a 3D model with a fine mesh contains a large number of elements and necessitates excessive simulation computing time, making parametric study infeasible. Two approaches can be used to overcome these drawbacks. The first is to build a 3D model to verify the 2D plane strain model, which can then be used to evaluate the packaging structure, and the second is to reduce the number of elements in the 3D model. In this work, we used multipoint constraint (MPC) technology to significantly reduce the number of elements. Panels (a) and (c) in Figure 9 illustrate 1/4 models of a simple WLP with full fine meshes, and panels (b) and (d) illustrate their MPC-reduced counterparts. This simple 1/4 WLP model will not take too much simulation time and was selected in this research to verify that the MPC model can obtain stress/strain results similar to the full fine mesh model.

After verifying the MPC model against the full model, the MPC model was used to simulate the test vehicle and the stacked PoP architecture. Figure 10 shows the 3D quarter symmetry finite element model. 

## 5. Results and Discussion

### 5.1. 2D Diagonal Plane Strain Model

After OBTCT, we inspected the failed test vehicle and found that the crack was located atop the solder joint with the largest DNP. Consistent with the experiment wherein we used glass substrate A, the simulation also showed that the maximum equivalent inelastic strain/stress was atop the outermost solder joint (Figure 11). In addition, the 248 life cycles simulated by Equation (4) are in good agreement with the experiment results of 249 mean cycles to failure (Figure 4). Thus, the failure prediction of the 2D model is in good agreement with the experiment data.

Next, we conducted a parametric study to optimize the reliability life under different combinations of upper and lower pad diameters. As evident in Figure 12, an upper pad larger than a lower pad resulted in better reliability. The Young’s modulus of silicon material is stronger than that of PCB, so a larger upper pad size is required to reduce the equivalent inelastic strain. A lower strain can have better reliability life, and the strain of the solder ball is related to the pad size, contact angle, and standoff height of the solder ball. In the simulation, when the upper pad diameter was 250 µm and the lower pad diameter was 180 µm, the best reliability life cycles could be achieved. That is, the pad on the interposer side should be larger than that on the PCB side. The optimal reliability life of 368 cycles was achieved at an upper pad/lower pad diameters ratio of approximately 1:0.72.

To meet condition G of the JEDEC standard, the thermal cycling life of the product must exceed 1000 cycles. To meet this standard with the optimized pad combination (upper pad 250 µm and lower pad 180 µm), we further optimized the SBL thickness and CTE of the glass substrate (Figure 13). Specifically, we simulated SBL thicknesses of 5, 10, and 20 µm and both substrate glasses A and B. Glasses A and B differ little in terms of Poisson’s ratio, but differ substantially in CTE (3.17 and 8.37 ppm/°C, respectively). Under the same conditions, the reliability life was higher for glass B than for glass A by 600 cycles on average, meaning that the reliability life is most sensitive to glass CTE. Moreover, a thicker SBL reduced solder stress/strain and thus favorably affected the thermal cycling reliability. Overall, the reliability life improved from 248 cycles to 1432 cycles with the following optimized combination: upper pad diameter, 250 µm; lower pad diameter, 180 µm; SBL thickness, 20 µm; glass CTE, 8.35 ppm/°C.

### 5.2. 3D Quarter Symmetry Model

The 3D FO-WLP PoP simulation model is much larger than the nonstacked FO-WLP, which makes 2D simulation the only feasible option for the parametric study of a stacked FO-WLP. To verify whether a 2D model can be applied to stacked packaging, a 3D model must first be established. To reduce the computing time, the number of elements and nodes must be decreased. First, we established a small WLP model (four solder balls per quarter model; Figure 9) both as a full fine-mesh model (86,656 elements; Figure 9a) and an MPC model (32,648 elements; Figure 9b). The corresponding computing times were 5 h 43 min and 2 h 13 min. We applied the same thermal loading as in the earlier analyses (−40 to 125 °C). Strain was found to be concentrated in a corner of the ball and the upper edge (Figure 14). After one to three thermal cycles, the strain increment in the two models differed by less than 1% (Figure 15). These simulation results demonstrate the feasibility of applying the MPC approach to simulate FO-WLP PoP with 112 solder balls per quarter model, as shown in Figure 10.

Next, we applied the design optimized in Section 5.1 (upper pad diameter, 250 µm; lower pad diameter, 180 µm; SBL thickness, 20 µm; glass CTE, 8.35 ppm/°C) to the 3D quarter symmetry model. Figure 16 shows the equivalent inelastic strain distribution in the FO-WLP. Strain accumulated in the solder ball with the maximum DNP, and the cycling life predicted by the 3D model differed from the 2D model prediction by only 1.3%.

We repeated the above simulation for the FO-WLP PoP structure. Again, we found that strain concentrated in the bottom layer of the solder ball with the maximum DNP, replicating the trend of failure at a corner location. Moreover, the maximum strain was in the upper pad (Figure 17). The upper and lower parts of the solder balls in the bottom layer were connected to the glass substrate and the PCB, respectively. Therefore, these solder balls could withstand greater stress than could solder balls in the upper layer. However, this double-layer structure reduced the cycling life to 974 cycles. In the PoP structure, the percentage of glass in this package increased; therefore, the equivalent inelastic strain due to CTE mismatch also increased, and a higher strain may lead to earlier failure.

Herein, the 2D and 3D models exhibited good consistency, with only 5% difference in reliability life. Therefore, we used the verified 2D models in the parametric analysis to optimize the design parameters in order to extend the reliability life of the FO-WLP PoP structure to more than 1000 cycles.

### 5.3. Parametric Analysis of FO-WLP PoP Structure Using the 2D Plane Strain Finite Element Model

In addition to the already optimized pad combination, three design parameters were considered in our parametric study: SBL thickness, chip thickness, and CTE of the glass substrate. In this parametric analysis, SBL thicknesses of 20, 25, and 30 μm, chip thicknesses of 90, 100, and 150 μm, and glass substrate CTEs of 8.35 and 9.8 ppm/°C were evaluated, yielding 16 combinations (Table 4). SBLs thicker than 30 μm were not evaluated due to the manufacturing difficulty. The results showed that the thicker the SBL, the higher the reliability and the higher the released stress/strain concentration. Furthermore, a lower chip thickness yielded a higher molding compound volume, which in turn increased the CTE of the whole package and thus narrowed the CTE mismatch between the package and PCB. Most importantly, glass CTE was found to have the strongest decreasing effect on the CTE mismatch between the glass substrate and the PCB. In summary, the reliability life of FO-WLP PoP can be increased by optimizing the SBL thickness and glass CTE (Figure 18). Of the 16 simulated combinations, combination 13—SBL thickness, 30 µm; chip thickness, 90 µm; glass substrate CTE, 9.8 ppm/°C—yielded the longest life cycle of 1427 cycles (Table 4). 

## 6. Conclusions

The simulation of 3D models is infeasible because of the high processing power and time requirements. In this study, to optimize the reliability life of FO-WLP and FO-WLP-type PoP under thermal cycling loading, 2D and 3D MPC technique finite element simulation models were established and experimentally verified to have similar accuracy. The reliability life of the verified 2D models was then optimized through parametric analysis. The CTE of the glass substrate was found to exert the strongest effect on reliability life. In addition, the upper and lower pad diameters and buffer layer thickness significantly affected the reliability life of the FO-WLP and FO-WLP-type PoP. Specifically, the reliability life of the solder joints was highest when the upper pad was larger than the lower pad (1:0.72). Moreover, a thicker buffer layer released a greater stress/strain concentration and thus positively affected the reliability life. However, SBLs thicker than 30 μm were not evaluated in this study due to the manufacturing difficulty. 

## Figures and Tables

**Figure 1 micromachines-12-00295-f001:**
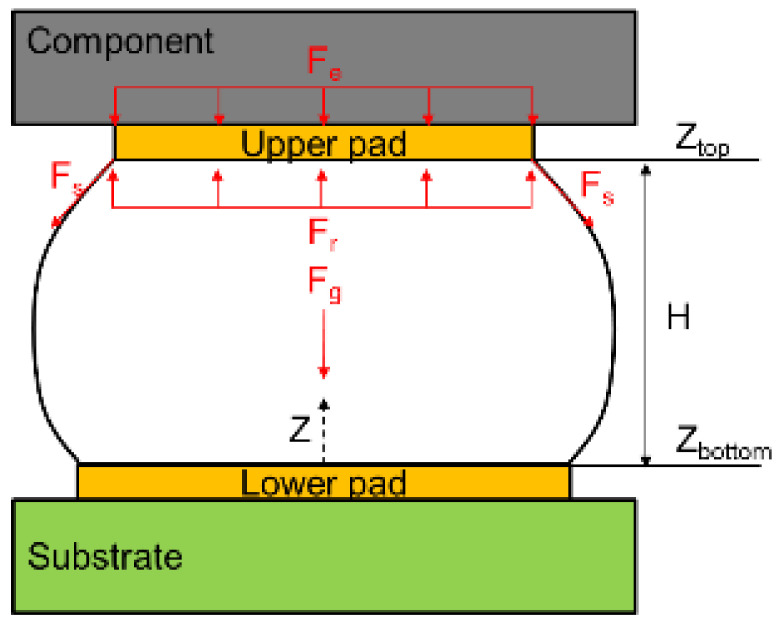
Geometry of the solder joints in the reflow process.

**Figure 2 micromachines-12-00295-f002:**
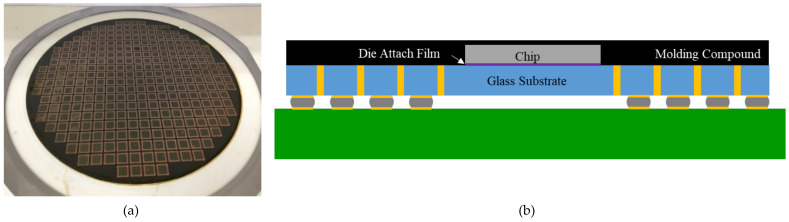
(**a**) Fan-out wafer-level package (FO-WLP) after debonding; (**b**) schematic of the fabricated FO-WLP onboard.

**Figure 3 micromachines-12-00295-f003:**
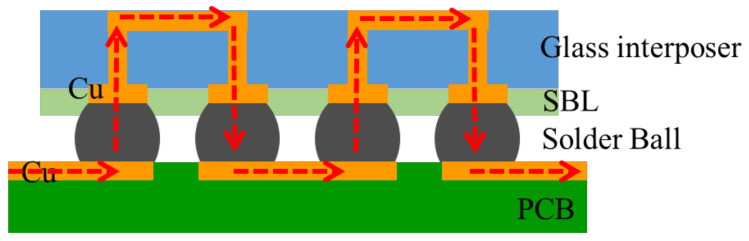
Schematic of the daisy chain. PCB, printed circuit board; SBL, stress buffer layer.

**Figure 4 micromachines-12-00295-f004:**
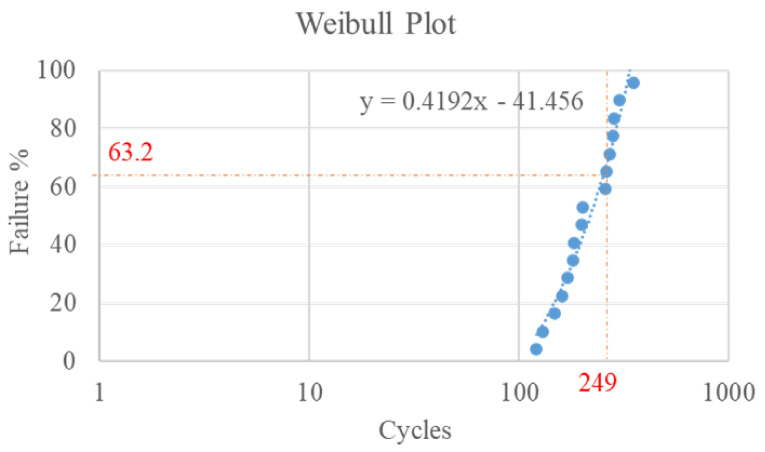
Weibull distribution of the glass interposer fan-out package.

**Figure 5 micromachines-12-00295-f005:**
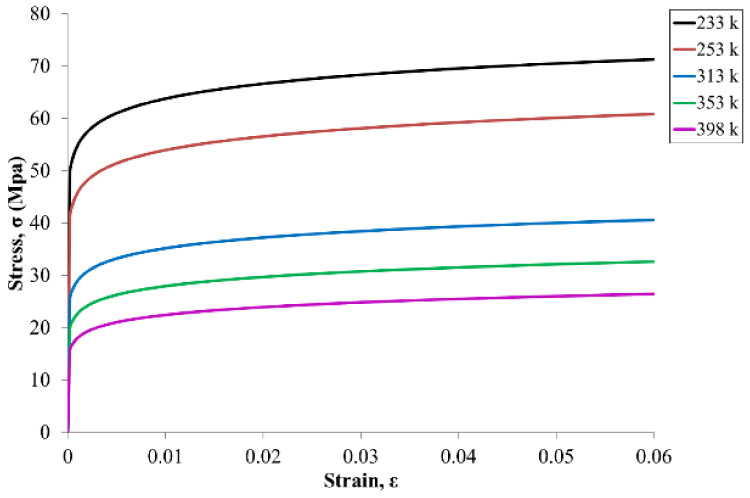
Stress–strain curves of Sn–3Ag–0.5Cu solder balls at different temperatures.

**Figure 6 micromachines-12-00295-f006:**
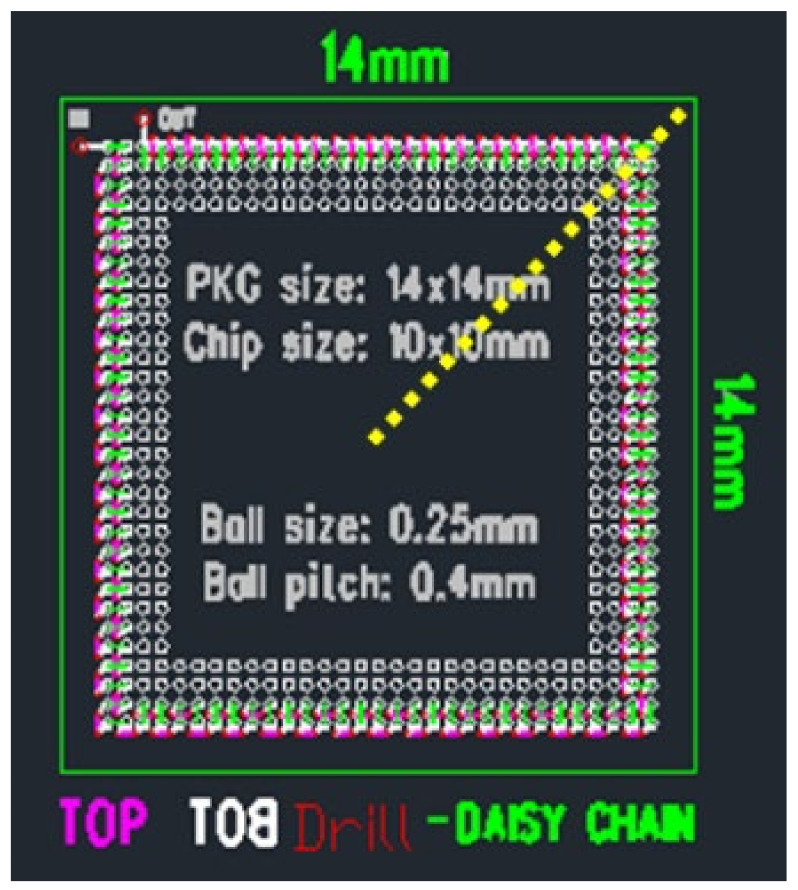
Top view of the test vehicle.

**Figure 7 micromachines-12-00295-f007:**
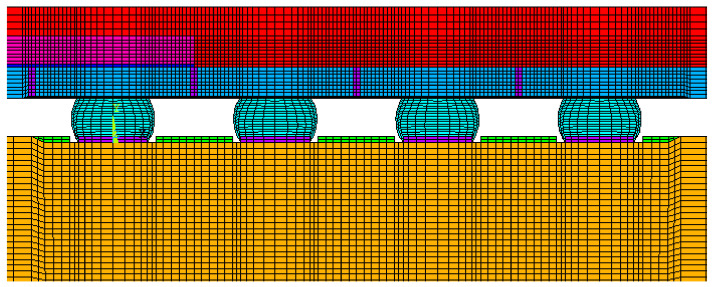
Two-dimensional (2D) finite element model of the FO-WLP.

**Figure 8 micromachines-12-00295-f008:**
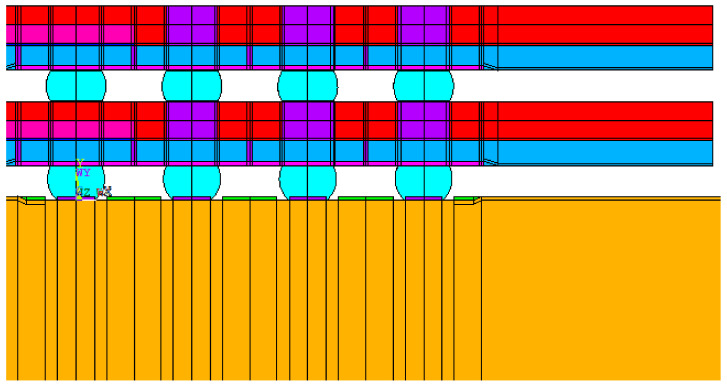
2D finite element model of the stacked FO-WLP (i.e., packaging-on-packaging (PoP)).

**Figure 9 micromachines-12-00295-f009:**
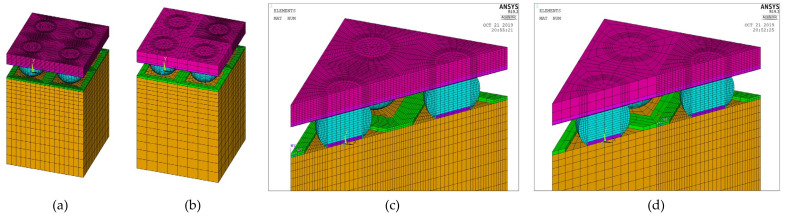
Three-dimensional (3D) finite element model: (**a**) Full fine-mesh model; (**b**) multipoint constraint (MPC) model; (**c**) cross-section view of the full fine-mesh model; (**d**) cross-section view of the MPC model.

**Figure 10 micromachines-12-00295-f010:**
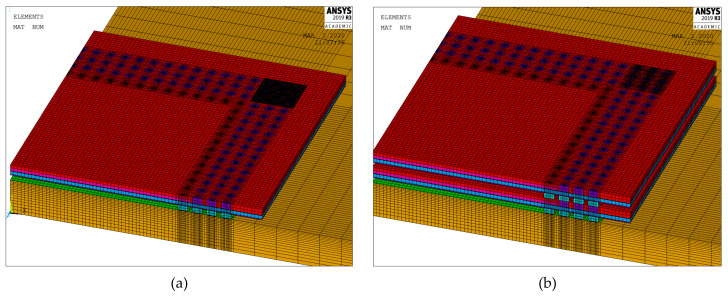
Three-dimensional MPC-reduced finite element models for the (**a**) FO-WLP and (**b**) FO-WLP PoP.

**Figure 11 micromachines-12-00295-f011:**
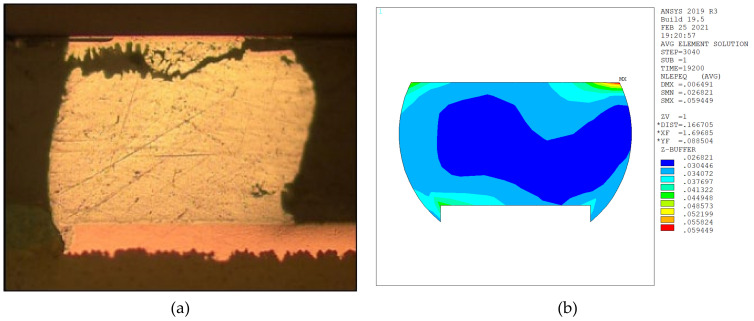
Failure point of solder joint after onboard thermal cycling testing (OBTCT): (**a**) Cross-section view; (**b**) equivalent inelastic strain distribution.

**Figure 12 micromachines-12-00295-f012:**
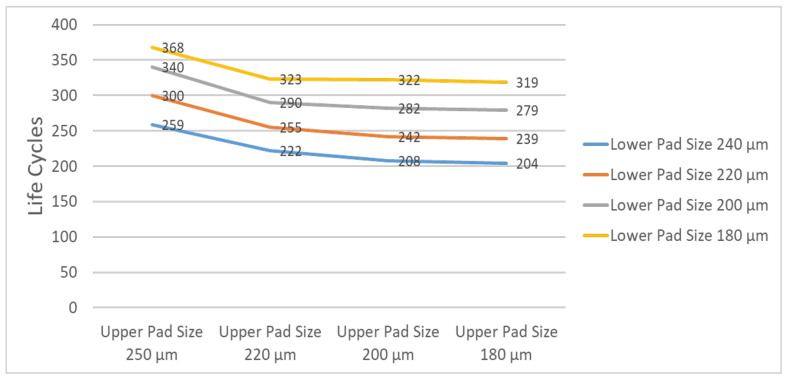
Simulated reliability life under different combinations of pad thicknesses.

**Figure 13 micromachines-12-00295-f013:**
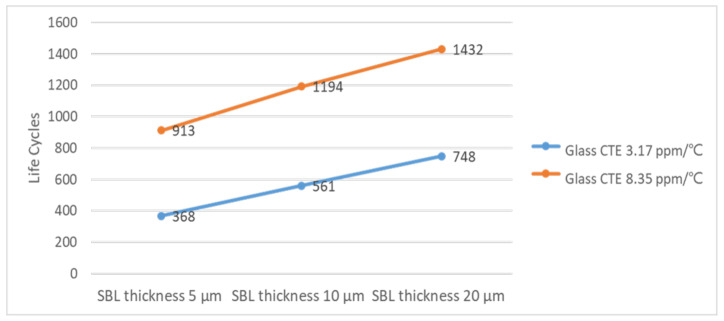
Simulated reliability life for stress buffer layers of different thicknesses.

**Figure 14 micromachines-12-00295-f014:**
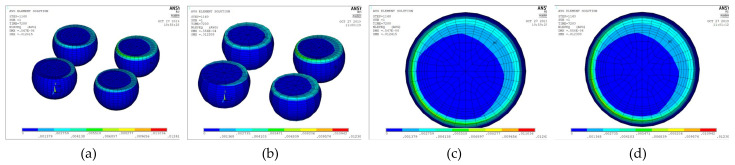
Equivalent inelastic strain distribution: (**a**) Full fine-mesh model; (**b**) MPC model; (**c**) maximum strain in the full model (top view); (**d**) maximum strain in the MPC model (top view).

**Figure 15 micromachines-12-00295-f015:**
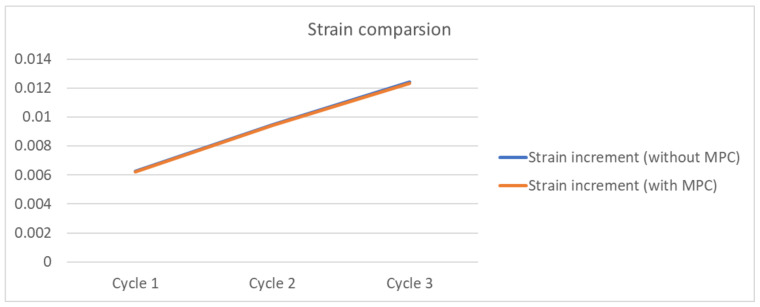
Strain increment in the full and MPC models.

**Figure 16 micromachines-12-00295-f016:**
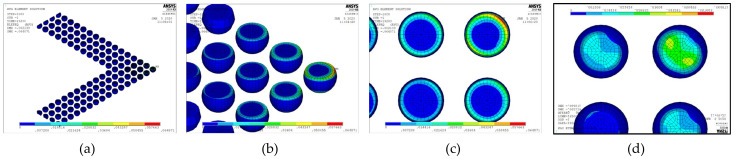
Equivalent inelastic strain distribution of the FO-WLP: (**a**) All solder joints; (**b**) location of the maximum strain; (**c**) top view of the bottom solder joint with the maximum distance from the neutral point (DNP); (**d**) bottom view of the solder joint with the maximum DNP.

**Figure 17 micromachines-12-00295-f017:**
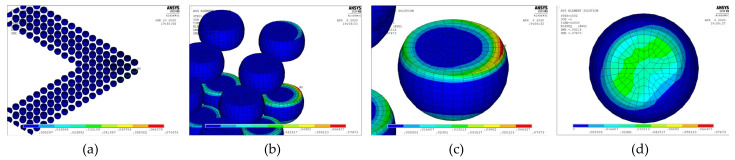
Equivalent inelastic strain distribution in FO-WLP PoP: (**a**) All solder joints; (**b**) location of the maximum strain; (**c**) top view of the solder joint with the maximum DNP; (**d**) bottom view of the solder joint with the maximum DNP.

**Figure 18 micromachines-12-00295-f018:**
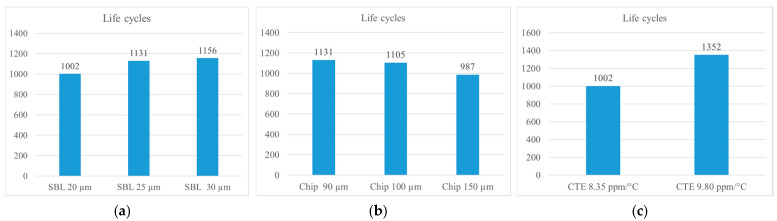
Life predictions of the FO-WLP PoP under different (**a**) SBL thicknesses, (**b**) chip thicknesses, and (**c**) CTEs.

**Table 1 micromachines-12-00295-t001:** Dimensions of the components in a fan-out wafer-level package (FO-WLP).

Component	Size (mm)
Glass substrate	14 × 14 × 0.10
Chip	10 × 10 × 0.10
Molding compound	0.19
Die attach film	10 × 10 × 0.01
Printed circuit board	77 × 132 × 1
Lower pad	0.24 × 0.02
Upper pad	0.25 × 0.002
Stress buffer layer (SBL; polyimide)	0.005
Through glass via	0.025 × 0.1
Solder ball diameter	0.25
Solder ball pitch	0.4

**Table 2 micromachines-12-00295-t002:** Summary of the material properties of the components in the FO-WLP. CTE, coefficient of thermal expansion.

Material	Young’s Modulus (GPa)	Poisson’s Ratio	CTE (ppm/°C)
Silicon	150	0.28	2.62
Stress buffer layer	2	0.33	55
Copper	68.9	0.34	16.7
SAC305 solder	Nonlinear/creep	0.4	22.36
Die attach film	1.66	0.26	17
Glass A	73.6	0.23	3.17
Glass B	71.7	0.21	8.37
Glass C	74	0.23	9.8
Printed circuit board	18.2	0.19	16
Molding compound	8.96	0.35	15

**Table 3 micromachines-12-00295-t003:** Constants in the creep equation [13].

Material	A (1/s)	B (1/MPa)	*n*	Q (J/mol)
**Value**	2631	0.0425	4.96	52,400

**Table 4 micromachines-12-00295-t004:** Combination of the design parameters in the parametric analysis of the FO-WLP PoP.

Item	SBL Thickness (µm)			Chip Thickness (µm)			CTE (ppm/°C)		Life Prediction
	20	25	30	90	100	150	8.35	9.8	
**1**	V			V			V		1002
**2**	V			V				V	1352
**3**	V				V		V		976
**4**	V				V			V	1337
**5**	V					V	V		923
**6**	V					V		V	1301
**7**		V		V			V		1131
**8**		V		V				V	1398
**9**		V			V		V		1105
**10**		V			V			V	1241
**11**		V				V	V		987
**12**			V	V			V		1156
**13**			V	V				V	1427
**14**			V		V		V		1129
**15**			V		V			V	1410
**16**			V			V	V		998

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
