# Peer review of "Reliability Evaluation of Fan-Out Type 3D Packaging-On-Packaging"

_micromachines, 2021, doi:10.3390/mi12030295_

Round 1
Reviewer 1 Report
Interesting work with a good approach for the study and comprehensive analysis.
Remarks and Questions:
Lines 55 – 64: It will be good to add an illustration that will include the parameters and the vectors directions for better understanding.
Line 120: ε and σ should be defined too.
Figure 10 b – is not clear, it is not seen what is written in the figure (the values of strains, etc.)
Line 166: Is there only one failure mode or there also other possible / observed failure modes?
Line 168: How the number of simulated life cycles was calculated? Using Eq. 4?
Lines 175 – 178: Why bigger pad on interposer side increase the reliability? Is there any explanation for upper pad : lower pad ratio?
Line 215: Why the FO-WLP PoP (double structure) reduce the number of thermal cycles regarding to the FO-WLP?
Figure 16 c – This is not a Top view.
Line 218: According to line 215, the number of cycles for FO-WLP PoP structure is below 1000 cycles.
What are the critical stresses and strains for the failure?
What are the stresses and strains before and after optimization (at failure cycle of the design before optimization)?
According to this study PoP structure reduce the number of cycles to failure. What is the limitation for number of layers?
Is the optimized design evaluated in thermal cycling? This information will be very good to verify the simulation results.
Optional:
109 – 111: Maybe better to move this paragraph to other place in the text?
Table 2 and Figure 4 to move after paragraph 113 – 117. Maybe it will be more convenient to read the text.

Author Response
1.Lines 55 – 64: It will be good to add an illustration that will include the parameters and the vectors directions for better understanding.
Reply: Thank you for your suggestion, we have added it in the article.
Figure 1. Geometry of solder joints in reflow process
2.Line 120: ε and σ should be defined too.
Reply: Thanks and the definition is update in Line 120. ε is strain, σ is stress.
3.Figure 10 b – is not clear, it is not seen what is written in the figure (the values of strains, etc.)
Reply: The figure 10(b) has been updated.
4.Line 166: Is there only one failure mode or there also other possible / observed failure modes?
Reply: According to the experimental results, all 16 samples failed at the top side of the outermost solder joint.
5.Line 168: How the number of simulated life cycles was calculated? Using Eq. 4?
Reply: Yes, we used the Coffin-Manson strain-based empirical model (Eq. 4) to evaluate the life cycle.
6.Lines 175 – 178: Why bigger pad on interposer side increase the reliability? Is there any explanation for upper pad : lower pad ratio?
Reply: The Young’s modulus of silicon material is stronger than that of PCB, so a larger upper pad size is required to reduce the equivalent inelastic strain. Lower strain can have better reliability life, and the strain of the solder ball is related to the pad size, contact angle and standoff height. In the simulation, the upper pad diameter is 250 um, and the lower pad diameter is 180 um can have the best reliability life.
7.Line 215: Why the FO-WLP PoP (double structure) reduce the number of thermal cycles regarding to the FO-WLP?
Reply: In the PoP structure, the percentage of glass in this package increases, therefore, the equivalent inelastic strain due to CTE mismatch will also increase, higher strain may lead to earlier failure.
8.Figure 16 c – This is not a Top view.
Reply: Change statement to “Top view of the bottom solder joint with the maximum DNP”
9.Line 218: According to line 215, the number of cycles for FO-WLP PoP structure is below 1000 cycles.
(1) What are the critical stresses and strains for the failure?
(2) What are the stresses and strains before and after optimization (at failure cycle of the design before optimization)?
(3) According to this study PoP structure reduce the number of cycles to failure. What is the limitation for number of layers?
(4) Is the optimized design evaluated in thermal cycling? This information will be very good to verify the simulation results.
Reply:
(1) Equivalent inelastic strain
(2) After optimization, the incremental inelastic strain at the top of the outermost solder joint was reduced from 2.08 x 10-2 to 7.89 x 10-3. Reliability increased from 249 cycles to 1,432 cycles.
(3) Based on our study, we have confidence to describe the 2 layers stacking can pass >1,000 cycles. We didn’t do simulation for the layer number more than 2
(4) Yes, the optimized structure is applied to the final product.
Reviewer 2 Report
In this manuscript, the reliability analysis of a glass-substrate type FOWLP was investigated by thermal cycling via FEM simulations. Three parameters, i.e. chip thickness, glass CTE and SBL thickness were varied and compared. Unfortunately, in my point of view, the paper is not organized properly and it is confusing for the audience to find out which parts are based on the experimental results and physical test vehicle and which part is based on pure simulations. Additionally, the novelty of this work is not clear, whereas the literature review should be improved. I encourage the authors to beef up the text, especially on the discussion part. The paper is short for a journal paper and the references are also not sufficient. My other comments are as follows:
- Please separate the experimental results from the simulation in two chapters. Describe which parts were fabricated and tested and which parts were only simulated.
- Add details of the die attach and SBL layer (material, type, manufacturer)
- Please discuss which kinds of glass were used or aimed to use. The question here is what are Glass A, B and C ?
- In Figure 2, the solder ball seems to be attached directly to Cu. Is it so or there are ENIG layer or other metallizations in between?
- In the introduction, please introduce SBL concept in FOWLP and its importance in one paragraph.
- In the proposed model, the reliability of TGV is missing? Please comment on that, since there can be a huge CTE mismatch between the glass and filled Cu in TGV.
- The plots should be graphically improved
Author Response
Reviewer: 2
In this manuscript, the reliability analysis of a glass-substrate type FOWLP was investigated by thermal cycling via FEM simulations. Three parameters, i.e. chip thickness, glass CTE and SBL thickness were varied and compared. Unfortunately, in my point of view, the paper is not organized properly and it is confusing for the audience to find out which parts are based on the experimental results and physical test vehicle and which part is based on pure simulations. Additionally, the novelty of this work is not clear, whereas the literature review should be improved. I encourage the authors to beef up the text, especially on the discussion part. The paper is short for a journal paper and the references are also not sufficient. My other comments are as follows:
1.Please separate the experimental results from the simulation in two chapters. Describe which parts were fabricated and tested and which parts were only simulated.
Reply: In chapter 3 (Test Vehicle Structure and Thermal Cycling), the test vehicle sample, dimensions and mean-cycle-to-failure are obtained through experiment. In chapter 4 and chapter 5 are obtained through simulation.
2.Add details of the die attach and SBL layer (material, type, manufacturer)
Reply: (1) Die attach film: Nitto, EM700
(2) SBL: JSR, Polyimide
- Please discuss which kinds of glass were used or aimed to use. The question here is what are Glass A, B and C ?
Reply: According to our study, glass CTE was found to have the highest influence on solder joint reliability life. CTE of glass A, B, and C are 3.17 ppm/°C, 8.35 ppm/°C, and 9.8 ppm/°C, respectively.
4.In Figure 2, the solder ball seems to be attached directly to Cu. Is it so or there are ENIG layer or other metallization in between?
Reply: ENIG is used on the Cu pad of PCB.
5.In the introduction, please introduce SBL concept in FOWLP and its importance in one paragraph.
Reply: SBL is widely used in package structure and this is a well-known mechanism that can be used to reduce solder strain. It can be found in many literatures.
- In the proposed model, the reliability of TGV is missing? Please comment on that, since there can be a huge CTE mismatch between the glass and filled Cu in TGV.
Reply: Yes, some TSV type package do have failure occur in cu via. In our study, during TCT reliability test, the daisy chain resistance is measured. According to the experimental results, all 16 samples failed at the top the outermost solder joint. TGV failed is not observed in our test vehicle, therefore, our simulation focuses on solder joint failure.
Round 2
Reviewer 2 Report
It can be accepted from my point of view.